# Galangin: A Promising Flavonoid for the Treatment of Rheumatoid Arthritis—Mechanisms, Evidence, and Therapeutic Potential

**DOI:** 10.3390/ph17070963

**Published:** 2024-07-19

**Authors:** Ghada Khawaja, Youmna El-Orfali, Aya Shoujaa, Sonia Abou Najem

**Affiliations:** 1Department of Biological Sciences, Faculty of Science, Beirut Arab University, Beirut 11-5020, Lebanon; 2Department of Experimental Pathology, Immunology and Microbiology, Faculty of Medicine, American University of Beirut, Beirut 11-0236, Lebanon; 3Health Sciences Division, Abu Dhabi Women’s College, Higher Colleges of Technology, Abu Dhabi P.O. Box 25026, United Arab Emirates; sabounajem@hct.ac.ae

**Keywords:** rheumatoid arthritis, galangin, anti-inflammatory, antioxidant, anti-arthritic

## Abstract

Rheumatoid Arthritis (RA) is a chronic autoimmune disease characterized by progressive joint inflammation and damage. Oxidative stress plays a critical role in the onset and progression of RA, significantly contributing to the disease’s symptoms. The complex nature of RA and the role of oxidative stress make it particularly challenging to treat effectively. This article presents a comprehensive review of RA’s development, progression, and the emergence of novel treatments, introducing Galangin (GAL), a natural flavonoid compound sourced from various plants, as a promising candidate. The bioactive properties of GAL, including its anti-inflammatory, antioxidant, and immunomodulatory effects, are discussed in detail. The review elucidates GAL’s mechanisms of action, focusing on its interactions with key targets such as inflammatory cytokines (e.g., TNF-α, IL-6), enzymes (e.g., SOD, MMPs), and signaling pathways (e.g., NF-κB, MAPK), which impact inflammatory responses, immune cell activation, and joint damage. The review also addresses the lack of comprehensive understanding of potential treatment options for RA, particularly in relation to the role of GAL as a therapeutic candidate. It highlights the need for further research and clinical studies to ascertain the effectiveness of GAL in RA treatment and to elucidate its mechanisms of action. Overall, this review provides valuable insights into the potential of GAL as a therapeutic option for RA, shedding light on its multifaceted pharmacological properties and mechanisms of action, while suggesting avenues for future research and clinical applications.

## 1. Introduction

### 1.1. Rheumatoid Arthritis (RA)

In rheumatoid arthritis, “arthr-” refers to joints, “-itis” means inflammation, and “rheumatoid” comes from rheumatism, which broadly refers to various musculoskeletal conditions. Therefore, rheumatoid arthritis (RA) is a chronic, inflammatory disorder that mostly affects the joints, but can also involve other organ systems like the skin and lungs as well [1,2]. RA is a chronic autoimmune disorder that impacts around 1% of the global population. While the disease can manifest at any age, it is most commonly diagnosed in people between the ages of 30 and 60 [3]. Interestingly, women are more than twice as likely to develop RA compared to men. This gender disparity may be attributed to the role of estrogen in modulating the immune response, potentially increasing women’s susceptibility to the disease [4,5].

#### 1.1.1. Development of RA

Rheumatoid arthritis is an autoimmune process that is typically triggered by an interaction between genetic and environmental factors [6]. A deficiency in a gene that encodes an immune protein, such as human leukocyte antigen (*HLA-DR1* and *HLA-DR4*), may result in rheumatoid arthritis when combined with environmental factors like cigarette smoke or specific infections [7,8,9]. These environmental factors can activate peptidyl arginine deiminases in mucosal cells, enhancing the post-translational conversion of arginine to citrulline in the presence of intracellular or matrix proteins, a process known as citrullination [10,11].

This process creates neo-epitopes, which are novel antigenic determinants formed from alterations in the protein sequence, that can be recognized by the adaptive immune system. These modified peptides are presented on major histocompatibility complex (MHC) proteins by antigen-presenting cells (APCs), activating T cells in lymphoid tissues. This, in turn, stimulates B cells to produce a variety of autoantibodies that recognize self-proteins, including rheumatoid factor (RF), which targets IgGs and anti-citrullinated protein antibodies (ACPAs), which target citrullinated proteins [12,13].

#### 1.1.2. Pathogenesis of RA

The presence of ACPAs and RF is associated with a more aggressive disease progression and can serve as a diagnostic and prognostic indicator [14,15]. Upon binding to their targets, these antibodies form immune complexes that accumulate in the synovial fluid. Subsequently, these complexes activate the complement system, which operates in an enzymatic cascade to facilitate joint inflammation and injury. Initially, the classical pathway is activated when the C1 complex binds to the Fc region of the antibodies in the immune complexes. This triggers a sequential activation of complement proteins C2, C4, and C3, ultimately forming the C3 convertase. The convertase cleaves C3 into C3a and C3b, where C3b binds to the immune complex, leading to the formation of the C5 convertase. This convertase cleaves C5 into C5a and C5b.

C5a acts as a potent anaphylatoxin, promoting inflammation, and C5b initiates the assembly of the membrane attack complex (MAC) composed of C5b, C6, C7, C8, and C9. The MAC creates pores in cell membranes, leading to cell lysis and contributing to tissue damage. Additionally, the activation of complement components C3a and C5a recruits and activates neutrophils, macrophages, and other immune cells to the site, further amplifying the inflammatory response and resulting in joint inflammation and injury [16,17].

In RA, T-helper cells and antibodies move into the bloodstream and travel to the joints [18,19]. Once there, T cells release cytokines like interferon-gamma (IFN-γ) and interleukin-17 (IL-17), which attract more inflammatory cells such as macrophages into the joint space [20,21]. Macrophages also produce inflammatory cytokines like tumor necrosis factor-alpha (TNF-α), interleukin-1 (IL-1), and interleukin-6 (IL-6), which, along with T cell cytokines, stimulate the growth of synovial cells [22,23]. This process leads to the formation of pannus, a thickened synovial membrane containing fibroblasts, myofibroblasts, and inflammatory cells [24]. Over time, the pannus can damage cartilage and other soft tissues and erode bone. Synovial cells also release enzymes that break down proteins in the cartilage [25,26]. Without protective cartilage, bones can rub against each other. Additionally, inflammatory cytokines increase the production of RANKL, a protein that allows T cells to bind to osteoclasts, triggering bone breakdown [27,28]. Chronic inflammation also promotes the formation of new blood vessels around the joint, bringing more inflammatory cells [29].

As the disease progresses, inflammation and damage occur in multiple joints on both sides of the body [30]. These inflammatory cytokines do not remain limited to the joints; instead, they spread through the bloodstream, affecting other organs. For instance, IL-1 or IL-6 can travel to the brain, causing fever [31]. When these cytokines reach the brain, they stimulate the production of prostaglandins, especially in the hypothalamus. Prostaglandins act on the hypothalamus, specifically in the preoptic area, which is the body’s thermoregulatory center. This release of prostaglandins leads to an increase in the hypothalamic set point for body temperature, causing the body to generate and retain heat, resulting in fever [32]. The systemic effects of these cytokines extend beyond fever. In muscles, they promote protein breakdown. In the skin and body organs, they lead to the formation of round collections of immune cells called rheumatoid nodules [33,34]. The liver responds by producing high amounts of hepcidin, a protein that decreases iron levels in the blood [35,36]. In the lungs, fibroblasts become overactive, leading to scar tissue formation that impairs gas exchange. The pleural cavities around the lungs can also become inflamed, filling with fluid (pleural effusion), which can affect lung expansion [37,38].

Hence, as Rheumatoid Arthritis progresses, the systemic spread of inflammatory cytokines leads to a wide range of extra-articular manifestations. These affect multiple organs and tissues throughout the body, resulting in diverse complications.

#### 1.1.3. Role of NF-κB, MAPK, JAK/STAT, and PI3K/Akt in the Pathogenesis of RA

RA is characterized by an intricate interplay of inflammatory pathways, including NF-κB, MAPK, JAK/STAT, and PI3K/Akt. All of these contribute to the chronic inflammatory state and joint damage observed in the disease [39,40]. The NF-κB pathway, a pivotal regulator of inflammation, is aberrantly activated in RA synovial cells, leading to the transcription of pro-inflammatory genes such as *TNF-α*, *IL-1β*, and *IL-6* [41]. These cytokines, in turn, activate the MAPK pathway, comprising p38 MAPK, ERK, and JNK subfamilies, which further promotes cytokine production and matrix metalloproteinase (MMP) expression, contributing to inflammation and tissue damage in RA [42].

Concurrently, the JAK/STAT pathway, activated by cytokines like IL-6 and IL-17 abundant in RA synovium, plays a crucial role in RA pathogenesis [43]. This pathway not only drives the differentiation of pro-inflammatory T-helper 17 (Th17) cells but also enhances the activation and recruitment of immune cells, including macrophages and neutrophils, to the inflamed synovium, perpetuating the inflammatory cascade. Additionally, the PI3K/Akt pathway, activated by growth factors and cytokines, promotes the survival and proliferation of synovial fibroblasts and immune cells, contributing to synovial hyperplasia and inflammation [44]. Akt activation also stimulates the expression of MMPs, notably MMP-9, further exacerbating tissue damage in RA joints.

Overall, the dysregulated activation of these signaling pathways in RA leads to a vicious cycle of inflammation, immune cell infiltration, and tissue destruction. Targeting these pathways individually or in combination presents a promising therapeutic approach for RA, aiming to alleviate inflammation, preserve joint function, and improve patient outcomes.

#### 1.1.4. Oxidative Stress in RA

The development of RA is closely associated with oxidative stress, a condition characterized by an imbalance between the excessive production of reactive oxygen species (ROS) and the body’s insufficient antioxidant defenses. This condition contributes to inflammation, immune dysregulation, and joint damage, all of which are characteristic of RA [45]. During the inflammatory process in RA, various cells such as neutrophils, macrophages, and synoviocytes produce ROS, including superoxide anions, hydrogen peroxide, and hydroxyl radicals [46]. These ROS are known to cause damage to lipids, proteins, and DNA, leading to cell dysfunction and death [47], and activate redox-sensitive signaling pathways like NF-κB, and MAPKs, which in turn promote the production of pro-inflammatory cytokines, chemokines, and MMPs. This chain of events further worsens the inflammation and joint damage experienced by RA patients [48,49,50]. The human body combats oxidative stress through a range of antioxidant mechanisms [51]. Enzymatic antioxidants, such as superoxide dismutase (SOD), catalase, and glutathione peroxidase, play a crucial role in detoxifying ROS [52]. Non-enzymatic antioxidants, including vitamins C and E, also help neutralize free radicals [53]. Additionally, the activation of transcription factors like NRF2 stimulates the production of antioxidant proteins, enhancing the body’s resilience to oxidative stress [54]. By understanding these defense systems, we can better appreciate the therapeutic potential of promising compounds, that not only reduce inflammation but also bolster these antioxidant defenses, thereby protecting joint tissues from oxidative damage and inflammation associated with RA.

Several studies have shown that RA patients have increased oxidative stress markers. For instance, the levels of lipid peroxidation products, such as malondialdehyde (MDA) and 4-hydroxynonenal (4-HNE), were observed to be higher in the synovial fluid and serum of RA patients [55]. Furthermore, the levels of antioxidants, such as superoxide dismutase (SOD), catalase (CAT), and glutathione peroxidase (GPx), were found to be decreased in RA patients, further exacerbating oxidative stress [56].

Additionally, three transcription factors—activator protein 1 (AP-1), hypoxia-inducible factor (HIF), and nuclear factor 2 (Nrf2)—play crucial roles in disease progression [57]. AP-1, a complex of proteins from the Fos and Jun families, regulates the expression of genes involved in cell proliferation, differentiation, and inflammation. In RA, AP-1 is activated by pro-inflammatory cytokines such as TNF-α and IL-1β, as well as by other stimuli like ROS and mechanical stress. Activated AP-1 in RA synovial fibroblasts promotes the expression of MMPs, contributing to the degradation of cartilage and bone [58]. Additionally, AP-1 regulates the expression of pro-inflammatory cytokines and chemokines, further amplifying the inflammatory response in RA.

In the hypoxic microenvironment of the RA synovium, HIFs play a critical role in regulating genes involved in angiogenesis, inflammation, and cell survival [59]. HIF-1α, a subunit of the HIF complex, stabilizes under hypoxic conditions and promotes the expression of vascular endothelial growth factor (VEGF) and other angiogenic factors, leading to excessive blood vessel formation (angiogenesis) in the synovium. This aberrant angiogenesis contributes to the chronic inflammatory state and pannus formation in RA. Moreover, HIF-1α regulates the expression of genes encoding pro-inflammatory cytokines and MMPs, further exacerbating joint inflammation and damage in RA.

Nrf2 regulates the expression of genes involved in antioxidant defense and detoxification of ROS. In RA, oxidative stress is a prominent feature, leading to increased ROS production and oxidative damage in the joints [60]. Nrf2 activation in response to oxidative stress is impaired in RA, contributing to the accumulation of ROS and oxidative damage. Restoring Nrf2 activity was proposed as a potential therapeutic strategy to alleviate oxidative stress and inflammation in RA.

In summary, AP-1 promotes joint inflammation and destruction by regulating the expression of MMPs and pro-inflammatory mediators. HIFs contribute to RA pathogenesis by promoting angiogenesis, inflammation, and cell survival in the hypoxic synovial microenvironment. Nrf2 dysfunction leads to increased oxidative stress and inflammation in RA joints. Therefore, chronic inflammation and oxidative stress are pivotal players in the aggravation of chronic inflammatory joint disease. In this regard, anti-inflammatory and antioxidant therapy may offer novel adjuvant/complementary treatment options aimed at enhancing disease control.

#### 1.1.5. Novel Treatments for RA

The ultimate goal of RA treatment is to reduce inflammation, minimize joint damage, and enhance the patient’s physical function and quality of life [61,62]. Treatment is expected to improve the clinical outcomes of RA by reducing synovitis, thereby decreasing the likelihood of further joint damage [63].

There are three general classes of drugs commonly used in RA therapy [61]: non-steroidal anti-inflammatory drugs or NSAIDs, corticosteroids, and disease-modifying antirheumatic drugs or DMARDs. Although NSAIDs and corticosteroids are effective at providing symptom relief for patients with RA, only DMARDs were demonstrated to modify the course of the disease and improve radiologic findings [64]. DMARDs are divided into two types: biological and synthetic small molecules (Figure 1) [65,66].

Biologics represent a novel approach to treating RA based on recent advances in understanding the immune and inflammatory processes associated with the disease [67,68]. TNF-α inhibitors were the first biological DMARDs approved for arthritis treatment [69]. They function by binding to TNF-α, thereby reducing its activity and minimizing inflammation. Blocking agents are another type of biological DMARD that interfere with the interaction between certain immune cells [70]. B-cell depleting agents are also used, reducing the number of B cells in the joints and lowering T cell activation and plasma cell count [71].

Synthetic small molecule DMARDs are also effective treatments for RA, acting by modifying the immune response within cells. Methotrexate (MTX), a non-biologic DMARD, is a commonly prescribed medication for RA treatment [72]. Initially developed as a cancer drug, MTX has proven beneficial in treating other inflammatory diseases like psoriasis [73]. MTX inhibits dihydrofolate reductase, an enzyme necessary for cell division and growth [74]. This disruption in nucleic acid production leads to decreased cell proliferation and suppression of the immune response. However, prolonged use of MTX can lead to toxicity in various tissues, including bone marrow suppression, liver damage, gastrointestinal irritation, pulmonary inflammation or fibrosis, and in high doses, kidney damage [75].

Many current treatments for RA can cause significant side effects, which may limit their long-term use. Treatment failure is also a concern, as some patients do not respond to existing therapies, leading to continued disease progression and joint damage. While many treatments can alleviate symptoms, achieving complete disease remission remains a challenge for numerous patients. With the development of new therapies, more options will become available to reduce inflammation in RA and potentially achieve disease remission. To avoid side effects and treatment failure, researchers are exploring natural sources of drugs, specifically bioactive compounds obtained from natural products.

Numerous studies were conducted to investigate the potential of natural alternatives for treating rheumatoid arthritis (RA) and preventing the side effects of Methotrexate (MTX), commonly used in RA treatment. Synthetic trans-Δ9-tetrahydrocannabinol (Δ9-THC) [76], Indole-3-Carbinol (I3C) found in cruciferous vegetables [77], an Aloe species’ leaf extract [78] Boswellia extract rich in Acetyl Keto Boswellic Acid (AKBA) [79], glabridin found in Glycyrrhiza glabra roots [80], and a natural agent named Silibinin [81] have shown anti-inflammatory and anti-arthritic effects. These natural agents were also shown to be effective in protecting the liver from MTX-induced damage. The combination therapy of these natural agents with MTX was found to be more effective in inhibiting arthritis and reducing MTX-induced hepatotoxicity in arthritic animals, highlighting the potential of natural compounds as adjuncts to conventional RA therapy.

### 1.2. Introduction of GAL as a Potential Therapeutic Agent and Its Sources

Flavonoids are polyphenols found in honey and various parts of plants, including barks, roots, rhizomes, stems, leaves, flowers, seeds, grains, fruits, and vegetables [82]. Flavonols, a subclass of flavonoids, possess a 3-hydroxyflavone backbone. Both flavonoids and flavonols exhibit a range of health benefits, including antioxidant, anti-inflammatory, and immune-modulatory properties, potentially offering relief from various ailments [83,84]. Galangin (GAL), a flavonol, is a promising contender that has demonstrated favorable outcomes in various disease models [85,86,87,88]. GAL is extracted from honey [89], propolis [90], and bee pollen [91,92], and is found in a wide variety of plants. Its sources are summarized in Table 1.

## 2. Bioactive Properties of GAL

### 2.1. Molecular Structure of GAL

GAL, a bioactive compound, is classified as a 3,5,7-trihydroxyflavone, a subclass of flavonoids. This molecule possesses a distinct chemical structure characterized by a chromone ring (also known as 1-benzopyran-4-one) substituted with hydroxyl groups at specific positions 3, 5, and 7. Additionally, a phenyl ring is linked to the chromone ring (see Figure 2) [117]. The molecular formula of GAL is C_15_H_10_O_5,_ and its molecular weight is 270.24 g/mol. A summary of the chemical and physical properties of galangin can be found in the Appendix A.

### 2.2. Pharmacological Properties of GAL

Using the ultra-high-performance liquid chromatography–tandem mass spectrometry (UHPLC–MS/MS) method, researchers studied the pharmacokinetic properties of GAL. They discovered that when GAL was administered orally or via intraperitoneal injection to a rat model, it was quickly absorbed (tmax = 0.25 h) and eliminated (t1/2 < 1.1 h) with an absolute bioavailability of 7.6%. GAL was found in high amounts in the kidney, liver, and spleen, and it was also able to cross the blood–brain barrier to reach the brain, although in lower concentrations [118]. GAL undergoes metabolism through glucuronidation reaction, producing metabolites such as galangin-3-O-β-D-glucuronic acid (GG-1) and galangin-7-O-β-D-glucuronic acid (GG-2), which were detected in the urine and plasma of rats [119,120].

Due to limitations in absorption and solubility that hinder its effectiveness, researchers have investigated methods to improve these properties of GAL for optimal delivery [121,122,123,124,125]. These methods include liposomes, which act as carriers to deliver GAL to targeted sites within the body [121]. Micelles, on the other hand, can significantly enhance the water solubility of GAL, making it more readily absorbed [122]. Additionally, nanostructured lipid carriers (NLCs) and nanoparticles (NPs) offer advantages like increased drug-loading capacity and controlled release [123,124]. Finally, β-cyclodextrin inclusion complexes can enhance both the solubility and stability of GAL, improving its therapeutic potential [125].

Given its interaction with CYP450 enzymes, GAL has the potential to alter the metabolism of other drugs. It was shown to inhibit several CYP450 enzymes, including CYP2A6, CYP2C8, CYP2C13, and CYP3A1, while activating CYP1A2 and CYP2B3 [126,127]. Therefore, caution should be exercised when co-administering GAL with other pharmacological agents. GAL administration, both in vitro and in vivo, exhibited no cytotoxicity on normal cells, suggesting a favorable safety profile [128,129]. Studies demonstrated good tolerance at concentrations up to 30 µM in microglial BV2 cells and at high oral doses of 320 mg/kg in rats [128,129]. Remarkably, this non-toxic effect on normal cells aligns with GAL’s ability to destroy cancer cells, suggesting potential avenues for further research into its therapeutic applications [130].

In vitro and in vivo studies have revealed that GAL possesses a broad spectrum of biological activities, including anti-inflammatory, antioxidant, antifibrotic, antihypertensive, antimicrobial, and anticancer properties [131,132,133,134,135]. These diverse pharmacological effects have led researchers to investigate its potential therapeutic applications in various disease models, as illustrated in Figure 3.

#### 2.2.1. Anti-Inflammatory Effects of GAL

GAL exhibited anti-inflammatory properties in various disease models, both in vivo and in vitro. These diseases include cardiovascular, renal, neural, skin, pulmonary, hepatic, gastrointestinal, pancreatic, foot, and retinal illnesses [86,132,136,137,138,139,140,141,142,143]. GAL works by modulating several signaling pathways that participate in the production of both proinflammatory and anti-inflammatory mediators, thereby alleviating inflammation. Detailed insights into these pathways are provided in Table 2.

Multiple studies in various disease models have consistently shown that GAL can inhibit the activation of NF-κB [136,137,138,140]. This inhibition occurs by targeting specific proteins in the NF-κB signaling pathway. These proteins, IκBα, IKKβ, and p65, undergo phosphorylation, a crucial step for NF-κB activation [146]. GAL inhibits this phosphorylation process, thereby preventing NF-κB from dissociating from its natural inhibitor, IκBα [146]. This effectively suppresses NF-κB activity, thereby hindering its ability to initiate the inflammatory cascade [146]. Consequently, GAL-mediated inhibition of NF-κB results in reduced levels of various pro-inflammatory cytokines, including TNF-α, IL-1β, and IL-6 [136,137,148,151,164]. Additionally, GAL reduces the expression of proinflammatory enzymes such as inducible nitric oxide synthase (iNOS) and cyclooxygenase-2 (COX-2) [132,136,137,142,148,149,156], which are responsible for the production of nitric oxide (NO) and prostaglandin E2 (PGE2), respectively, both contributing to inflammation [129,149]. Furthermore, GAL modulates NF-κB by regulating other factors such as Toll-like receptor 4 (TLR4) and Peroxisome proliferator-activated receptor gamma (PPARγ) [139,140,151,153,160,165]. In studies using lipopolysaccharide-stimulated microglia models and in vivo models of pulmonary inflammation in vivo study, GAL demonstrated its ability to activate PPARγ, a molecule that downregulates NF-κB and inflammatory cytokine production [151,160]. GAL also disrupted the TLR4-NF-κB p65 signaling axis, further reducing inflammation in lipopolysaccharide (LPS)-injured rat intestinal epithelial (IEC-6) cells and sulphate sodium-induced ulcerative colitis in mice [140,165].

GAL’s influence extends beyond NF-κB. It exerts its anti-inflammatory effects through additional pathways, including the PI3K/AKT, MAPK, and NOD-like receptor family pyrin domain containing three (NLRP3 inflammasome) pathways [144,145,146,151,152,153]. GAL may also help control MMPs, enzymes that contribute to inflammation in RA as discussed previously. Notably, GAL inhibits MMP expression in activated microglia, human neuroblastoma cells, and an animal model of induced pulmonary inflammation [151,152,157]. Furthermore, in lipopolysaccharide-induced neuroinflammation in rats, GAL inhibited the Dipeptidyl peptidase 4 (DPP-4) activity, ultimately increasing levels of the anti-inflammatory molecule GLP-1 [153]. Furthermore, GAL activates Nrf2, a molecule known to be compromised in RA. This activation of Nrf2, as demonstrated in various in vivo models of liver, gastrointestinal, and skin inflammation, leads to the restoration of the Nrf2/HO-1 pathway and the production of the potent anti-inflammatory enzyme HO-1 [156,161,162,164].

Overall, GAL’s multifaceted anti-inflammatory actions, including NF-κB inhibition, cytokine reduction, MMP control, and Nrf2 activation, position it as a promising therapy for RA. By addressing inflammation, immune dysfunction, and joint degradation, GAL offers a comprehensive approach to combating RA’s complex pathogenesis.

#### 2.2.2. Antioxidant Effect of GAL

Due to its structure, particularly the presence of hydroxyl groups, GAL (3,5,7-trihydroxyflavone) effectively functions as an antioxidant. This functionality arises from its ability to readily donate a hydrogen atom, specifically from its 3-hydroxyl group. This donation transforms a portion of the GAL molecule into a 3-flavonoid phenoxyl radical. The phenoxyl radical acts as a free radical scavenger, neutralizing free radicals [166].

Endogenous antioxidants, including superoxide dismutase (SOD), catalase (CAT), and glutathione peroxidase (GPx), serve as the primary shield against cellular damage caused by reactive oxygen species (ROS). These enzymes facilitate the conversion of ROS into less reactive molecules. SOD specifically catalyzes the dismutation of superoxide radicals into hydrogen peroxide, a less harmful species. Furthermore, CAT offers additional protection by catalyzing the reduction of hydrogen peroxide into water and oxygen. Reduced glutathione (GSH) scavenges free radicals and acts as a substrate for several detoxifying enzymes. Therefore, a compromised antioxidant defense system can lead to the breakdown of hydrogen peroxide into highly reactive hydroxyl radicals, which can disrupt various cellular components [167].

Numerous studies have shown that GAL exhibits protective effects against oxidative stress. These studies revealed that GAL upregulated the activity of various antioxidant enzymes and molecules, including SOD, CAT, glutathione S-transferase (GST), GPx, and GSH [128,140,141,144,145,156,167,168,169,170,171,172,173]. Additionally, it increased the levels of antioxidant molecules like ascorbic acid (vitamin C) and α-tocopherol (vitamin E) [128,168,170]. Moreover, GAL suppressed lipid peroxidation, leading to a reduction in thiobarbituric acid reactive substances (TBARS) and MDA levels [133,140,144,145,156,167,170,173]. GAL’s ability to reduce ROS levels [144,174] may also be explained by its effect on NADPH oxidase-1 (NOX-1). This enzyme is a major source of ROS production in cells, and studies using a neurotoxicity-induced rat model have shown that GAL treatment downregulates NOX-1 [175].

Additionally, GAL administration inhibited protein oxidation and nitration, resulting in a decrease in protein carbonyls (PCO) and nitrotyrosine levels across various disease models. These models include induced nephrotoxicity, fructose feeding to induce metabolic syndrome, and a hepatorenal disease model [144,170,173]. It also downregulated Acyl-CoA Synthetase Long-Chain Family Member 4 (ACSL4) and 4-Hydroxynonenal (4-HNE) levels, attenuating oxidative stress in AR42J pancreatic cells [141].

Finally, GAL treatment restored and upregulated the Nrf2/HO-1 pathway, leading to an increase in heme oxygenase-1 (HO-1) expression. HO-1 is a potent antioxidant enzyme that helps eliminate harmful free radicals and alleviate oxidative stress. This, in turn, protects dermal fibroblasts from senescence following H_2_O_2_ exposure and mitigates the severity of acute pancreatitis, psoriasis-like skin inflammation, and cognitive alterations in rodent models [131,141,156,175].

Therefore, GAL’s antioxidant potential is derived from its direct scavenging of free radicals and reinforcement of the cellular antioxidant defense system. By enhancing the activity of antioxidant enzymes and molecules, it effectively shields cells from oxidative stress damage. This multifaceted antioxidant activity holds significant promise as oxidative stress mediators are pivotal in the development of joint damage in RA. Given the direct contribution of oxidative stress to the progression of RA, there is a compelling rationale to further explore GAL as a potential therapeutic strategy for safeguarding joints and managing RA.

#### 2.2.3. Immunomodulatory Effect of GAL

The immunomodulatory effects of GAL were shown in several in vivo and in vitro studies, as summarized in Table 3.

These effects contribute to a reduced inflammatory response through multiple mechanisms. First, GAL suppresses the infiltration of inflammatory cells, including neutrophils, lymphocytes, macrophages, eosinophils, and mast cells [154,158,178]. Notably, mast cells are key mediators of inflammatory reactions, releasing signaling molecules like histamine that cause redness and swelling (edema) [154]. Interestingly, GAL treatment directly suppresses histamine release in human mast cells (HMC)-1 by decreasing intracellular calcium levels [177]. Furthermore, GAL’s anti-inflammatory properties extend to its effects on macrophages. In LPS-stimulated RAW 264.7 macrophages, GAL downregulates COX-2, an enzyme crucial for the production of inflammatory prostaglandins [94]. This finding aligns with an in vivo study where GAL treatment attenuated myocardial ischemia-reperfusion injury in mice. This protective effect was associated with an increase in anti-inflammatory M2 macrophages and a decrease in proinflammatory M1 macrophages following GAL administration [181].

GAL’s anti-inflammatory effects extend beyond just suppressing immune cell infiltration. It also acts by downregulating key signaling pathways within the very cells it targets. This included inhibiting the activation of JNK, p38, ERK MAPKs, and NF-κB pathways in LPS-stimulated RAW 264.7 macrophages and human mast cells (HMC)-1, all of which are known contributors to the production of proinflammatory cytokines [94,177]. Furthermore, GAL disrupts the maturation process of these proinflammatory cytokines in HMC-1 by reducing caspase-1 activity [177]. Similarly, in LPS-stimulated RAW 264.7 macrophages, GAL targets upstream mediators in inflammatory signaling [94]. It inhibits interleukin-1 receptor-associated kinase 1 (IRAK-1), a key player in TLR signaling that triggers proinflammatory cytokine production [94]. Additionally, GAL modulates the JAK/STAT pathway by suppressing JAK-1, another crucial component of proinflammatory cytokine signaling within these macrophages [94]. This multi-pronged approach is evident in the reduced levels of pro-inflammatory mediators observed in various models. Treatment with GAL in LPS-stimulated macrophages and a concanavalin A-induced hepatitis model significantly decreased the production of cytokines (TNF-α, IFN-γ, and IL-12), chemokines (CXCL10 and MIP-1α), and adhesion molecules (ICAM-1) [94,178].

Beyond its influence on innate immune cells like macrophages and mast cells, GAL also modulates the adaptive immune response by affecting T lymphocytes. Following treatment with GAL in the bleomycin-induced pulmonary fibrosis model, atopic dermatitis model, and LPS-induced dendritic cell model, T cell activation and expansion were reduced. These studies revealed that GAL downregulated CD8+ T cells and CD4+ T cells (Th1, Th2, and Th17 cells), and the levels of cytokines associated with both Th1 (IFN-γ) and Th2 (IL-4, IL-5, IL-13, IL-31, and IL-32) immune responses [154,179,180]. GAL further dampens the inflammatory response by suppressing the production of immunoglobulins involved in allergic and inflammatory responses [154,158]. In models of ovalbumin-induced airway inflammation and atopic dermatitis, GAL treatment inhibited the production of immunoglobulin E (IgE) and decreased serum levels of IgG2a [154,158].

GAL’s influence extends to antigen-presenting cells. Research has investigated the potential role of GAL’s immunomodulatory effect on dendritic cells (DCs) [180]. GAL treatment of LPS-stimulated DCs promoted the development of tolerogenic dendritic cells (tolDCs) [180]. Compared to bone marrow-derived DCs (BMDCs), GAL-treated DCs (Gal-DCs) showed lower levels of CD86, a costimulatory molecule, and reduced expression of major histocompatibility complex class II (MHC-II) molecules, indicating diminished antigen presentation capabilities. Gal-DCs displayed an increase in the production of the anti-inflammatory cytokine IL-10 and programmed death ligand 1 (PD-L1) expression, which was associated with the activation of MAPKs such as ERK, JNK, and p38. Moreover, Gal-DCs induced allogeneic CD4 T cells differentiation into regulatory T cells [180]. This suggests that GAL skews DC towards a tolerogenic phenotype, leading to suppressed inflammatory T cell responses.

In conclusion, GAL’s immunomodulatory effects align well with the pathogenesis of rheumatoid arthritis (RA), making it a promising treatment option. GAL’s ability to suppress inflammatory cell infiltration, modulate macrophage and mast cell activity, and reduce proinflammatory cytokine production directly addresses these key aspects of RA pathogenesis. Additionally, its effects on T lymphocytes and antigen-presenting cells suggest a broader impact on the adaptive immune response, which is often dysregulated in RA. By targeting multiple components of the immune response implicated in RA, GAL offers a comprehensive approach to mitigating inflammation and potentially slowing down joint damage in RA patients.

#### 2.2.4. GAL and Autoimmunity

In the context of autoimmune diseases, GAL was shown to inhibit the activation of immune complex-stimulated neutrophils by FcγRs (Fc gamma receptors) and CRs (complement receptors). This effect reduces the activity of myeloperoxidase and horseradish peroxidase enzymes, which are involved in the production of reactive oxygen species (ROS) by neutrophils [176]. Consequently, GAL helps to reduce the overall ROS production, potentially mitigating tissue damage associated with autoimmune conditions.

Building on its ability to modulate immune cell activity and ROS production, a study using an experimental autoimmune encephalomyelitis (EAE) mouse model showed that GAL treatment ameliorated demyelination, inhibited the infiltration of mononuclear cells (MNCs) into the spinal cord, and reduced T cell proliferation and differentiation. Notably, GAL led to a decrease in TH1 and TH17 cells in the spinal cords of EAE mice. Furthermore, GAL treatment impaired the function of dendritic cells (DCs), including their ability to present antigens and produce cytokines (IL-6, IL-12, and IL-23). These combined effects contributed to the alleviation of clinical symptoms of EAE [87]. This suggests that GAL holds promise as a therapeutic strategy for multiple sclerosis and other neuroinflammatory diseases.

GAL’s therapeutic potential extends beyond neuroinflammatory conditions to other autoimmune diseases. In a study on psoriasis, an autoimmune skin disease causing chronic inflammation, GAL demonstrated its antioxidant and anti-inflammatory properties in an induced psoriasis-like skin inflammation model. This effect was mediated by the recovery of the Nrf2/HO-1 pathway and the upregulation of antioxidant enzymes and markers, including SOD, CAT, GST, GSH, glutathione reductase (GR), and vitamin C [156]. These findings suggest that GAL may offer therapeutic benefits for various autoimmune diseases by alleviating inflammation and oxidative stress.

While GAL exhibits therapeutic potential, it is important to consider its compatibility with existing medications used for autoimmune diseases. MTX is a well-established medication for managing autoimmune diseases such as rheumatoid arthritis and psoriatic dermatomyositis. Like all drugs, MTX has several side effects including the development of hepatotoxicity. GAL exerted its hepatoprotective effects in a rat model of MTX-induced hepatotoxicity by suppressing inflammation through the downregulation of NF-κB p65, iNOS, and the proinflammatory cytokines TNF-α, IL-1β, and IL-6. Furthermore, GAL alleviated oxidative stress induced by MTX by reducing the levels of ROS, NO, and MDA, and by upregulating antioxidant enzymes and molecules (GSH, SOD, and CAT), thereby mitigating MTX-induced hepatotoxicity [86]. The therapeutic potential of combining MTX with GAL for rheumatoid arthritis remains unexplored. Further research is warranted to investigate their efficacy and safety in this context.

## 3. Mechanism of Action of GAL against Arthritis: In Vivo and In Vitro Studies

GAL was studied for its potential therapeutic properties against bone disorders. Its anti-proliferative properties were shown to inhibit the growth of cancer cells and promote apoptosis in osteosarcoma [182]. GAL also exhibits immune protective effects by enhancing the immune system’s response to infections and reducing inflammation. These properties make GAL a promising candidate for the treatment of various bone disorders, such as osteoporosis and rheumatoid arthritis.

### 3.1. Effect of GAL on Osteoarthritis (OA)

Osteoarthritis (OA) is a prevalent form of arthritis characterized by joint degeneration [183]. Several studies suggest that GAL could be a potential treatment for OA, as shown in Table 4.

In vivo, GAL was found to prevent cartilage destruction, slow down the development and progression of OA, and protect articular cartilage [184,185,186,187,188,189]. In vitro, studies have demonstrated that treatment with GAL can decrease the expression of catabolic factors in chondrocytes, ameliorate the loss of extracellular matrix (ECM) components, and promote osteogenic differentiation [184,185,186,188,189,190]. Mechanistically, Huang et al. identified GAL as a potent substance with remarkable anti-inflammatory properties. It effectively reverses the inflammatory response caused by exposure to IL-1β in rat chondrocytes, reduces the degradation of collagen II and aggrecan, and inhibits Akt phosphorylation and NF-κB activation [185]. Treatment with GAL significantly reduces levels of ROS, lipid peroxidation IL-1β, IL-6, and TNF-a, while increasing levels of CAT, SOD, Gpx, and GSH.

Su et al. also observed that GAL significantly reduces the mRNA and protein expression levels of C-terminal cross-linked telopeptide of type II (CTX-II) [187]. Furthermore, Wang et al. showed that GAL has the ability to suppress the inflammatory response that is induced by IL-1β and ameliorate African Caribbean Leukaemia Trust (ACLT)-induced cartilage degeneration by effectively inhibiting the NF-κB pathway, as well as the JNK and ERK pathways [186]. Lin et al. demonstrated that GAL prevents ECM degradation by activating Proline and Arginine Rich End Leucine Rich Repeat Protein (PRELP) and inhibiting the PI3K-AKT signaling pathway. This helps in safeguarding chondrocytes from oxidative stress by inhibiting the expression of ROS and MDA while promoting the expression of SOD and CAT [188]. Moreover, Li et al. found that GAL suppresses RANKL-induced ERK-MAPK, p38-MAPK, and NF-κB signaling pathways in bone marrow macrophages by attenuating phosphor-ERK, phosphor-p38, and phosphor-p65, while stabilizing IκB-α expression [190]. Recently, Zeng et al. explored the effects of GAL on autophagy and osteogenesis in bone marrow stromal cells (BMSCs). The results showed that GAL enhanced the PKA/CREB signaling pathway in BMSCs, inducing autophagy and increasing osteogenic differentiation [189]. This suggests that GAL directly inhibits osteoclastogenesis via the NF-κB and MAPK signaling pathways, offering a potential avenue for preventing bone loss associated with OA.

### 3.2. Effect of GAL on Rheumatoid Arthritis (RA)

The antioxidant and immunomodulatory properties of GAL in RA were investigated in several studies [191,192,193,194]. GAL was found to prevent osteoclastic bone destruction and inhibit osteoclastogenesis by reducing RANKL-induced JNK, p38, and NF-κB activation in osteoclast precursors and in mice with collagen-induced arthritis [191]. Santos et al. demonstrated that GAL effectively reduces the production of ROS in neutrophils and inhibits myeloperoxidase (MPO) activity [194]. Furthermore, GAL can inhibit the expression of inflammatory mediators such as IL-1β, TNF-α, IL-18, PGE2, and NO in a dose-dependent manner in RA fibroblast-like synovial cells (RAFSCs), as well as the expression of iNOS and COX-2. Additionally, GAL can increase the activity of SOD and decrease MDA content in a dose-dependent manner [192]. Fu et al. discovered that GAL could suppress pro-inflammatory signaling in synoviocytes through the inhibition of the NF-κB/NLRP3 pathway, making it a potential therapeutic agent for RA [192]. Finally, GAL can also suppress inflammation, cell proliferation, migration, and invasion, while promoting apoptosis in rheumatoid arthritis fibroblast-like synoviocytes (RAFLSs) by modulating the PI3K/AKT pathway [193]. All these studies are summarized in Table 5.

## 4. Conclusions and Future Perspectives

The encouraging findings from the mentioned studies suggest that GAL, a natural flavonoid compound found in various plants such as Alpinia officinarum and Alpinia galanga, holds potential as a therapeutic agent for treating rheumatoid arthritis (RA). The search strategy involved querying PubMed using MeSH terms, covering publications from 1985 to 2024. GAL was shown to possess significant anti-inflammatory and antioxidative properties, which could be beneficial in reducing the inflammation and joint damage associated with RA. The compound functions by inhibiting the production of inflammatory mediators and reducing oxidative stress within the joints.

However, further research is necessary to determine the synergistic effects of GAL compared to the first-line drug for treating RA, such as methotrexate. Additionally, its potential additive and protective effects against the side effects of conventional RA drugs need a thorough evaluation. Understanding the precise mechanisms of action and potential interactions with other medications is crucial.

Clinical studies are also required to assess the safety and efficacy of GAL in humans before considering its widespread use. This involves investigating the optimal dosage, the potential adverse effects, and understanding how GAL may interact with other medications commonly used in RA treatment.

If proven effective and safe, GAL could provide a natural alternative to conventional anti-inflammatory drugs, such as NSAIDs and DMARDs, which often have side effects with prolonged use. Moreover, these study findings could pave the way for the development of other natural compounds with similar therapeutic potential, offering new avenues for the treatment of inflammatory diseases beyond RA.

## Figures and Tables

**Figure 1 pharmaceuticals-17-00963-f001:**
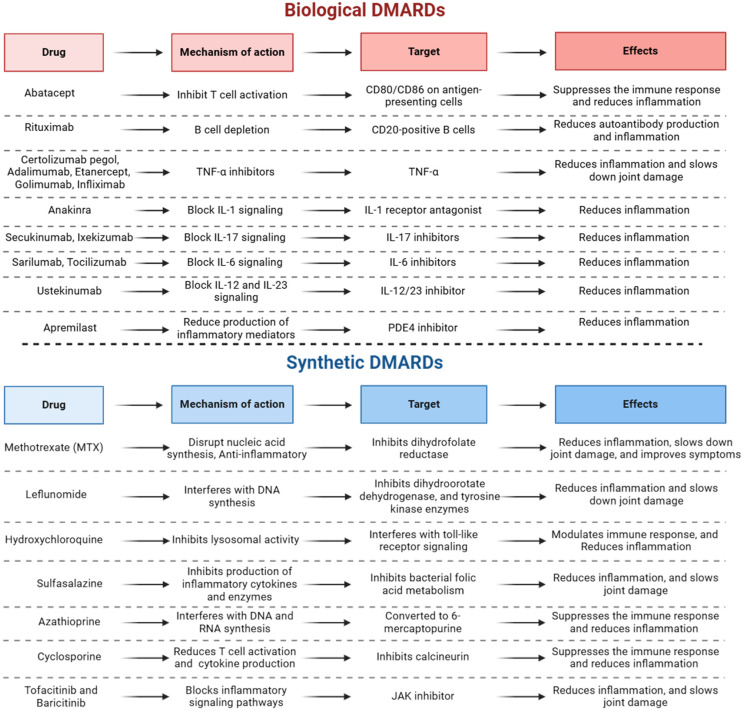
Overview of DMARDs used in RA treatment: mechanisms and effects.

**Figure 2 pharmaceuticals-17-00963-f002:**
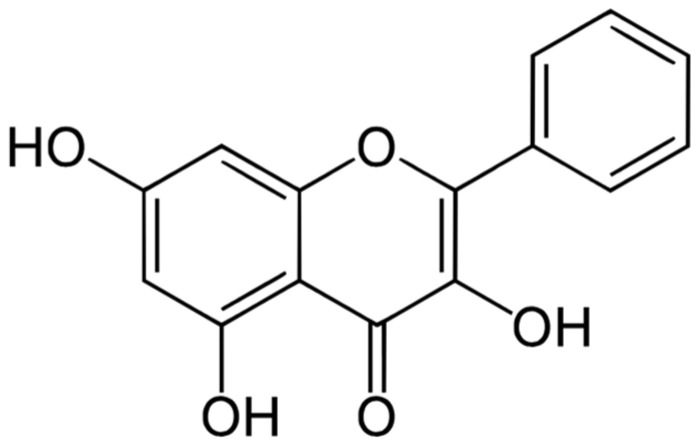
Chemical structure of GAL.

**Figure 3 pharmaceuticals-17-00963-f003:**
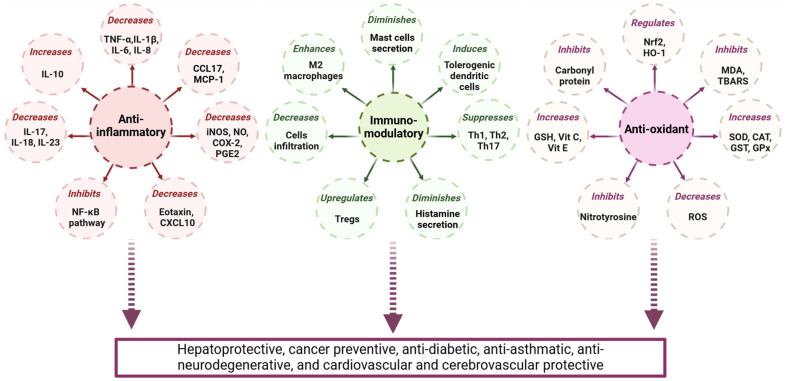
Different biological activities and numerous health benefits of GAL.

**Table 1 pharmaceuticals-17-00963-t001:** Sources of GAL.

Category	Source	References
Honeybee Products	Propolis, Bee Pollen	[89,90,91,92]
Plants (Rhizomes)	Alpinia officinarum, Alpinia calcarata, Alpinia galanga	[93,94,95]
Plants (Roots)	Coleus vettiveroides	[96]
Plants (Leaves)	Mexican Oregano (Lippia graveolens), Nothofagus gunnii, Eugenia catharinensis, Castanea sativa, Piper aleyreanum, Japanese Alnus sieboldiana	[97,98,99,100,101,102,103]
Plants (Buds)	Poplar (Populus nigra)	[104,105,106,107]
Plants (Other)	Poplar tree gum, Fruits (Sechium hybrid, Sechium edule Perla Negra, Campomanesia xanthocarpa, Prunus cerasus (Oblacinska), Psidium cattleianum), Plantain peel, Jujube peel and seed, Crocus sativus flower petals	[108,109,110,111,112,113,114,115]
Marine Fungus	Chaetomium globosum	[116]

**Table 2 pharmaceuticals-17-00963-t002:** Anti-inflammatory mechanisms of GAL.

Type of Inflammation	GAL’s Mechanism of Action	Model	References
Renal	Decreased NF-κB p65, iNOS, and TNF-α, IL-1β, IL-6 levels	Rat model of diabetic nephropathy	[136]
	Inhibited ERK and p38 MAPK signaling and NF-ĸB activation and the release of proinflammatory cytokines	Rodent model of nephrotoxicity	[144,145]
	Suppressed PI3K/AKT, NF-κB, NLRP3 inflammasome, and the production of TNF-α, IL-1β, PGE2, and NO	Rat kidney epithelial cells (NRK-52E)	[146]
Cardiovascular	Suppressed *TNF-α*, *IL-6*, *NF-κB*, *COX-2*, and *iNOS* gene expression	Albino Wistar rat model of cardiac inflammation and fibrosis	[132]
	Decreased VCAM-1, TNF-R1, TNF-α, and the activity of NF-κB	Rat model of hypertension	[133]
	Decreased NF-κB p65 activation and inhibited MEK1/2-ERK1/2-GATA4 and PI3K/AKT-GSK3β pathways	Mouse model of aortic banding mediated pressure overload and angiotensin II treated H9c2 cells	[147]
	Decreased NF-κB p65, iNOS, and TNF-α, IL-1β, IL-6 levels	Rat model of diabetic cardiomyopathy	[148]
	Positively regulated adiponectin/AdipoR1 signaling pathway and COX-2 and reduced NF-κB activity	Rat model of cardiometabolic syndrome	[149]
	Downregulated renin-angiotensin system (RAS) and IL-6/TNF-α and upregulated endothelial nitric oxide synthase (eNOS) pathway	Rat model of cardiometabolic syndrome	[150]
Neural	Reduced prostaglandin E2	Pentylenetetrazole (PTZ) mouse epilepsy model	[102]
	Suppressed production of proinflammatory mediators (TNF-α, IL-1β, IL-6, COX-2, iNOS) and inhibited phosphorylation of JNK, p38, AKT, and NF-κB p65	Rat model of Parkinson’s disease	[137]
	Suppressed phosphorylation of p38 MAPK, JNK, PI3K/Akt, and NF-κB, activated PPARγ, reduced proinflammatory mediators (TNF-α, IL-6, MMP-3, MMP-9, iNOS), and increased anti-inflammatory mediators (IL-10, HO-1)	Lipopolysaccharide (LPS) injected mouse brains and LPS-stimulated microglial cell line (BV-2 cells)	[151]
	Suppressed MMP-9 expression and inhibited phosphorylation of multiple signaling molecules including c-Src, Pyk2, PKC isoforms, Akt, mTOR, MAPKs (ERK1/2, JNK1/2, p38), FoxO1, c-Jun, and p65	Human neuroblastoma cell line (SK-N-SH cells)	[152]
	Inhibited proinflammatory mediators (iNOS, IL-1β, NO) and suppressed phosphorylation of JNK, p38 MAPK, and NF-κB	LPS-stimulated microglial cell line (BV-2 cells)	[129]
	Reduced proinflammatory cytokines (IL-8, TNF-α), inhibited TLR4/NLRP3 inflammasome and DPP-4, and increased GLP-1 levels	Rat model of LPS-induced neuroinflammation	[153]
Skin	Attenuated NF-κB p65 activation and proinflammatory cytokine release (TNF-α, IL-1β, IL-6)	Human HS68 dermal fibroblasts	[138]
	Inhibited ERK, p38 MAPKs, and NF-κB pathways and suppressed TNF-α/IFN-γ-induced production of CCL17 and proinflammatory cytokines (IL-6, TNF-α, IL-1β)	Mouse model of atopic dermatitis and human keratinocytes (HaCaT cells)	[154]
	Attenuated NF-κB/p65 activation, inhibited Erk1/2 and JNK activation, suppressed production of NO, iNOS, and IL-6, and downregulated serum IgE levels	Mouse model of atopic dermatitis and RAW264.7 macrophages	[155]
	Activated Nrf2/HO-1 pathway, inhibited NF-ĸB pathway, downregulated proinflammatory mediators (COX-2, iNOS) and cytokines (IL-17, IL-23, IL-1β), and upregulated anti-inflammatory cytokine IL-10	BALB/c mice model of induced psoriasis-like skin inflammation	[156]
Pulmonary	Inhibited collagen deposition, lowered alpha-smooth muscle actin, disrupted TGF-β1 pathway, and reduced VEGF and MMP-9	Induced chronic asthma mice model and airway smooth muscle cell culture (ASMC)	[157]
	Activated PPARγ, reduced IL-4, IL-5, IL-13, TNF-α, IL-17, NO, eosinophil peroxidase, and IgE, and increased IFN-γ	Induced allergic asthma mice model	[139]
	Suppressed activation of NF-κB p65 and TNF-α-mediated translocation, decreased production of inflammatory mediators including monocyte chemoattractant protein-1, eotaxin, CXCL10, and VCAM-1, IL-4, IL-5, IL-13, and iNOS	Mouse model of induced airway inflammation	[158]
	Inhibited NF-κB activation and reduced TNF-α and IL-6 levels	Mouse model of induced acute lung injury	[159]
	Activated PPARγ and DNMT3A, leading to suppressed ERK, p65, and AP-1, and reduced proinflammatory cytokines (IL-6, TNF-α)	Rat model of induced inflammatory lung injury	[160]
Hepatic	Suppressed NF-κB p65, iNOS, and proinflammatory cytokines (TNF-α, IL-1β, IL-6)	Rat model of induced hepatotoxicity	[86]
	Induced Nrf2/HO-1 pathway, elevated PPARγ, inhibited NF-κB activation, and decreased proinflammatory cytokines (TNF-α, IL-1β)	Rat model of induced hepatotoxicity	[161,162]
Gastrointestinal	Reduced TLR4 expression, decreased HMGB1, inhibited NF-κB p65 activation, and decreased proinflammatory cytokines (IL-6, TNF-α)	Mouse model of induced ulcerative colitis	[140]
	Downregulated IL-8 secretion	Human gastric adenocarcinoma (AGS) cells	[163]
	Activated the Nrf2/HO-1 pathway and suppressed the NF-κB pathway and its downstream proinflammatory mediators (COX-2, iNOS, and TNF-α/IL-6)	Mouse model of induced ulcerative colitis	[164]
	Suppressed TLR4/NF-κB signaling pathway, increased anti-inflammatory cytokines (IL-10, TGF-β), and decreased proinflammatory cytokines (IL-1β, IL-6, TNF-α), and PGE2	LPS-injured rat intestinal epithelial (IEC-6) cells	[165]
Pancreatic	Activated Nrf2/HO-1 pathway and inhibited NF-κB p65 and proinflammatory mediators, such as TNF-α, IL-18, MCP-1, and CXCL10	Mouse model of pancreatitis	[141]
Foot	Suppressed the expression of NF-κB, TNF-α, COX-2, and PGE2	Acute inflammation and pain rat model	[142]
Retinal	Inhibited the ERK1/2-NF-κB/Egr1 pathway and proinflammatory cytokines (TNFα, IL-1β, and IL-6)	D-glucose-induced microglia BV2 cells and retinal inflammatory injury in diabetic mice model	[143]

**Table 3 pharmaceuticals-17-00963-t003:** GAL’s immunomodulatory mechanisms.

Condition/Cell Type	Mechanism of Action	Key Findings	Reference
Experimental Autoimmune Encephalomyelitis (EAE) Model	Inhibits mononuclear cell infiltration, reduces T cell proliferation and differentiation (Th1 and Th17 cells), and impairs dendritic cell function	Alleviated autoimmune encephalomyelitis	[87]
LPS-stimulated RAW 264.7 Macrophage Cells	Downregulates COX-2, reduces NO and proinflammatory cytokines (TNF-α, IL-1β, IL-6), and inhibits IRAK-1, JAK/STAT pathway, MAPK (p38 and ERK), and NF-κB	Reduced inflammation	[94]
Atopic Dermatitis (AD) Model	Reduces infiltration of inflammatory cells (eosinophils, mast cells), decreases histamine levels, inhibits Th1 and Th2 cytokines, and reduces serum IgE and IgG2a	Reduced inflammation	[154]
Ovalbumin-(OVA-) Induced Airway Inflammation Model	Reduced total leukocytes (eosinophils, neutrophils, and lymphocytes) and downregulated the production of IgE	Alleviate induced airway hyperresponsiveness and inflammation	[158]
Neutrophils	Inhibits FcγRs and CRs activation, and reduces myeloperoxidase and horseradish peroxidase activity	Decreased ROS production, reduced tissue damage	[176]
Human Mast Cells (HMC)-1	Downregulates JNK, p38, and NF-κB pathways, suppresses histamine release and reduces activation of Caspase-1	Reduced inflammation	[177]
Concanavalin A (ConA)-induced Hepatitis (CIH) Model	Suppresses infiltration of inflammatory cells (neutrophils, macrophages, T cells), inhibits T cell activation via STAT1 pathway, and reduces proinflammatory cytokines and chemokines	Reduced inflammation	[178]
Bleomycin-Induced Pulmonary Fibrosis Model	Reduced the number of CD4+ and CD8+ T cells and dendritic cells	Alleviate pulmonary fibrosis	[179]
Dendritic Cells (DCs)	Promotes tolerogenic DCs (tolDCs) and stimulates regulatory T cells (Tregs)	Skews DCs towards tolerogenic phenotype and suppresses inflammatory T cell responses	[180]
Myocardial Ischemia-Reperfusion Injury	Induces autophagic flux via PI3K/AKT/mTOR pathway, increases anti-inflammatory M2 macrophages, and decreases proinflammatory M1 macrophages	Reduced inflammation	[181]

**Table 4 pharmaceuticals-17-00963-t004:** A synopsis of the in vitro and in vivo studies assessing GAL’s impact on OA.

	Model and GAL’s Dose	Effect on Disease Progression	Signaling Pathway and Antioxidant Defense	Reference
In vivo	Rat osteoblasts (ROB) 10^−8^ g/mL to 10^−4^ g/mL for 24 h	Promote osteoblast calcification Decrease ALP activity	N/D	[184]
	Anterior Cruciate Ligament Transection (ACLT) in SD rats 0.1 mL/day for 8 weeks (5 mg/mL)	Decrease OARSI score	N/D	[185,186]
	Monoiodoacetate (MIA) in SD rats 10 or 100 mg/kg/day for 14 days	Reduce CTX-II, IL-1b, IL-6, and TNF-a levels	Disturbance of pro-oxidant/antioxidant. Reduce ROS and lipid peroxidation levels Increase catalase, SOD, Gpx, and GSH levels	[187]
	Destabilization of the medial meniscus (DMM) in SD rats 20, 40 and 60 mg/kg, twice per week for 4 weeks	Decrease OARSI score Decrease MMP13 and ADAMTS5 Increase the levels of COL2A1 and ACAN	N/D	[188]
	Glucocorticoid-induced osteoporosis (GIOP) in C57BL/6 mice 10 or 40 mg/mL/day for 8 weeks	Alleviate bone damage	PKA/CREB-mediated autophagy signaling. Increase in LC3B and p-CREB levels	[189]
In vitro	IL-1β (10 ng/mL) treated chondrocytes Primary mouse bone marrow macrophages (BMMs) [5–20] µM for 24 h	Decrease iNOS and COX-2 expression Decrease MMP1, MMP3, MMP13 and ADAMTS5 Attenuate collagen II and aggrecan degradation	Inhibit NF-κB, MAPK and PI3K/AKT pathways Inhibit phosphorylation of ERK, JNK Akt, IKKαβ, IKBα and P65	[185,186,190]
	Human OA primary chondrocytes [0–100] µM for 24 h	Decrease MMP13 and ADAMTS5 Increase the levels of COL2A1 and ACAN	Inhibit PI3K-AKT pathway Downstream oxidative stress Inhibit phosphorylation of PI3K and AKT Activate PRELP Inhibit the expression of ROS and MDA Promote the expression of SOD and CAT	[188]
	Dexamethasone-treated human Bone marrow-derived mesenchymal stem cells (BMSCs) [5–20] µM for 24 h	Promote osteogenic differentiation Increase protein expression levels of Runx2, OCN, OPN	PKA/CREB, AKT/mTOR, PI3K/AKT and Wnt/β-catenin Upregulate p-PKA/PKA and p-CREB/CREB Downregulate p-mTOR/mTOR	[189]

**Table 5 pharmaceuticals-17-00963-t005:** A synopsis of the in vitro and in vivo studies assessing GAL’s impact on RA.

	Model and GAL’s Dose	Effect on Disease Progression	Signaling Pathway and Antioxidant Defense	Reference
In vivo	Collagen-induced arthritis (CIA) in DBA/1J mice 10, 50, or 100 mg/kg for 25 days	Reduce arthritis score and paw edema Improve bone/cartilage destruction, synovial hyperplasia, and pannus formation Reduce production of IL-1b, TNF-a, IL-17 and RANKL	N/D	[175]
	Collagen-induced arthritis (CIA) in SD rats 10, 20, and 40 µg for 4 weeks	Improve bone/cartilage destruction, synovial hyperplasia, and pannus formation Improve histological score Increase body weight Decrease TNF-α, IL-1β, and IL-6 levels	Downregulate PI3K/AKT/mTOR signaling pathway Inhibit of pPI3K, pAKT, and pmTOR protein levels	[178]
In vitro	Co-cultures of bone marrow-derived macrophages and primary osteoblasts 1, 10, and 20 mg/mL	Inhibit osteoclast formation Decrease TNF-α, IL-1β, and IL-17 levels	Suppress phospho-JNK and phospho p38 MAPK Inhibit NF-kB/p65 and IkBa phosphorylation level Increase IkBa level	[175]
	Human neutrophil-10 µM	Suppress O_2_^−^, ROS and MPO	N/D	[176]
	Primary human RA fibroblast-like synovium cells (RAFSCs) 1, 5, and 10 ng/mL for 24 h	Decrease IL-1β, TNF-α, IL-18, PGE2 and NO levels Inhibit iNOS and COX-2 expression levels	Downstream oxidative stress Suppress NF-κB/NLRP3 signaling pathway Decrease SOD activity and increase MDA content Decrease ASC, pro-caspase-1/caspase-1, p-IκBα, p-NF-κB, IL-1β and NLRP3 expression	[177]
	Rheumatoid arthritis fibroblast-like synoviocytes (RAFLSs) [10–160] µM for 24, 48, and 72 h	Suppress inflammation, proliferation, migration and invasion Promote apoptosis of RAFLSs	N/D	[178]

## Data Availability

Not applicable.

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
