# Peer review of "Galangin: A Promising Flavonoid for the Treatment of Rheumatoid Arthritis—Mechanisms, Evidence, and Therapeutic Potential"

_pharmaceuticals, 2024, doi:10.3390/ph17070963_

Round 1

Reviewer 1 Report

Comments and Suggestions for Authors

Khawaja et al., in their review article, described the mechanisms and therapeutic potential of Galangin and suggested this molecule to be a promising flavonoid for the treatment of Rheumatoid Arthritis. Although the review is interesting and provides evidence, certain essential information is missing. Also, several paragraphs need to be elaborated and clarified. My comments are below:

 ·       23-24: First, describe the key gaps this review is addressing, then what new knowledge is being contributed by this review.

·       Line 50: define neo-epitopes

·       60-61: This should be expanded a bit to cover the exact mechanisms and players involved in complement activation.

·       79: How did traveling to the brain cause fever? Can you elaborate on the pyogenic mechanisms?

·       171-188: It's superficial. The authors should describe the mechanisms behind the activity of DMARDs. I suggest to add a figure explaining the targets of different drugs

·       189-192: Needs clarity. Why are new therapies needed (identify gaps first), and then what is the direction of the development of new treatments?

·       211-212: What disease models? How are they related to RA?

·       Fig 2 is hard to read. The font size must be increased. I suggest replacing it with a table or changing the figure to add biological activities, pathways, and specific target molecules/proteins/genes, etc.  

·       Table 2: Add a column to show studies were in vitro (then name which cell line model) or animal or human (mention which model was used for each)

·       315-325: Gal administration where? What models were used?

·       Table 4 and 5: why are there empty cells? Also, the font should be consistent

·       Need to add details about Gal’s dose, properties, toxicity and pharmacokinetics

Comments on the Quality of English Language

 Minor editing of English language required

Reviewer 2 Report

Comments and Suggestions for Authors

Dear Authors,

Please add the search strategy including date and database to search.

 You have discussed the positive effects of Galangin in RA, but there are negative results for example increased MDA in reference NO 177 that you have not discussed.

 In Table NO 5, please add a column for methods, duration, dose, sample size for in vivo studies, and also separate in vitro studies.

 Differentiating your tables into "effect on disease progression" and "effect on signaling pathway" has problems, for example; increased MDA and decreased SOD activity should not be included under "effect on signaling pathway".

Please add abbreviations below the tables.

 With respect,

Reviewer 3 Report

Comments and Suggestions for Authors

The manuscript by Khawaja et al. summarized information on galangin in the treatment of RA. The authors presented previous findings on the mechanisms and effects of galangin. The manuscript was well structured and comprehensive. However, similar reviews are available. The other specific comments are as follows.

1. Some similar reviews are available, such as “Galangin: A food-derived flavonoid with therapeutic potential against a wide spectrum of diseases” (https://doi.org/10.1002/ptr.8013). What is the novelty and contribution of this review? What is new in this review compared with previous reviews?

2. Please avoid bolding the main text.

3. Figure 2 should be revised since the texts embedded are unclear. If unnecessary, please avoid using too many colors.

4. All the tables should follow the journal guidelines.

Comments on the Quality of English Language

Minor editing of English language required

Reviewer 4 Report

Comments and Suggestions for Authors

This was an interesting review of a topic of current interest based on the extensive and proper use of literature data. Please find my comments below in order to improve the quality of the article prior to its publication.

General comments:

-The titles of the Tables must be above the Tables (not below). Please correct it.

-I would like you to display in Figures the main mechanisms of action regarding the development of RA as well as the mechanism of action of galangin.

Specific comments

-Lines 113-115: Describe better the terminology of oxidative stress. Antioxidant defenses are associated with a response of the human body but this is not clearly highlighted in the text.

-Figure 2: The size of the letters in Figure 2 is too small and the text cannot be read unless you make a high zoom. Please improve the image in order to be readable at 100% zoom.

Comments on the Quality of English Language

The use of the English language was fine. Minor corrections need to take place.

Round 2

Reviewer 1 Report

Comments and Suggestions for Authors

Thank you for the revisions. The manuscript looks much better now. 

Reviewer 3 Report

Comments and Suggestions for Authors

The manuscript was appropriately revised and can be accepted.